# Influence of Terbium Ions and Their Concentration on the Photoluminescence Properties of Hydroxyapatite for Biomedical Applications

**DOI:** 10.3390/nano11092442

**Published:** 2021-09-19

**Authors:** Andrei Viorel Paduraru, Ovidiu Oprea, Adina Magdalena Musuc, Bogdan Stefan Vasile, Florin Iordache, Ecaterina Andronescu

**Affiliations:** 1Department of Science and Engineering of Oxide Materials and Nanomaterials, Faculty of Applied Chemistry and Materials Science University Politehnica of Bucharest, 060042 Bucharest, Romania; andrei93.paduraru@yahoo.com (A.V.P.); ovidiu73@yahoo.com (O.O.); amusuc@icf.ro (A.M.M.); bogdan.vasile@upb.ro (B.S.V.); 2National Centre for Micro and Nanomaterials, University Politehnica of Bucharest, 060042 Bucharest, Romania; 3“Ilie Murgulescu” Institute of Physical Chemistry, Romanian Academy, 060021 Bucharest, Romania; 4National Research Center for Food Safety, University Politehnica of Bucharest, 060042 Bucharest, Romania; 5Department of Biochemistry, Faculty of Veterinary Medicine, University of Agronomic Science and Veterinary Medicine, 011464 Bucharest, Romania; floriniordache84@yahoo.com; 6Academy of Romanian Scientists, 50085 Bucharest, Romania

**Keywords:** terbium, rare earth ions, doped hydroxyapatite, coprecipitation method, photoluminescence, biological system imaging

## Abstract

A new generation of biomaterials with terbium-doped hydroxyapatite was obtained using a coprecipitation method. The synthesis of new materials with luminescent properties represents a challenging but important contribution due to their potential applications in biomedical science. The main objective of this study was to revel the influence of terbium ions on the design and structure of hydroxyapatite. Different concentrations of terbium, described by the chemical formula Ca_10−x_Tb_x_(PO_4_)_6_(OH)_2_, where x is in the range of 0 to 1, were considered. The consequence of ion concentration on hydroxyapatite morphology was also investigated. The morphology and structure, as well as the optical properties, of the obtained nanomaterials were characterized using X-ray powder diffraction analysis (XRD), Fourier Transform Infrared spectrometry (FTIR), SEM and TEM microscopy, UV-Vis and photoluminescence spectroscopies. The measurements revealed that terbium ions were integrated into the structure of hydroxyapatite within certain compositional limits. The biocompatibility and cytotoxicity of the obtained powders evaluated using MTT assay, oxidative stress assessment and fluorescent microscopy revealed the ability of the synthesized nanomaterials to be used for biological system imaging.

## 1. Introduction

Photoluminescence is a significant and valuable instrument for the in situ study of tissue engineering and restoration, with fluorescent molecules having been used in clinical trials for a long time. In the last decade, many researchers have been focused on developing new biological luminescent compounds with special properties, e.g., high quantum yield and long fluorescence lifetime, for use in the medical field [1,2]. In this context, many bioceramics have been selected and used for reconstruction and/or to repair different types of tissue, applied as a coating, as cement, or as nanoparticles [3].

Hydroxyapatite (HAp, Ca_10_(PO_4_)_6_(OH)_2_), the essential mineral constituent of hard biological tissues such as teeth and bones [4], is used as a synthetic cement for bone and dental reconstruction due to its osteogenic, osteoconductive and osteoinductive properties [5,6]. Numerous research studies have suggested that nanoparticles of calcium phosphate can be used as fluorescent samples after doping with lanthanide by substitution of the Ca^2+^ ions, a method that also enhances the biological properties of HAp [7,8,9,10,11]. The substitution of Ca^2+^ ions can be made using various doping agents: carbon compounds [12,13,14], fluorine, chlorides and silicate [12] or cations—monovalent (K^+^, Ag^+^, Na^+^) [15,16,17], divalent (Zn^2+^, Sr^2+^, Mg^2+^, Cd^2+^, Cu^2+^) [18,19,20] and trivalent (Fe^3+^, Al^3+^, rare earths ions) [21,22]. In addition to medical uses, cation-substituted hydroxyapatite can also be used in multiple applications: sensors, catalysts, water decontamination, optoelectronics or radionuclides, and heavy metal remediation [23].

Luminescent rare earths have a surprising spectral nature, which allows them to be successfully used as a non-isotopic substitute for organic fluorophore compounds. Aside from in vivo detection of cell function applications, this group of elements is widely used for the luminescent marking of molecules to elucidate the structure and function of proteins and enzymes [24]. The photoluminescent properties of rare earth ions are mostly due to the narrow and strong electronic interconfigurational *f*-*f* emission transitions [25], a large effective Stokes shifts, and high quantum yields [26].

Terbium (Tb, atomic number: 65) is an important element in the rare earth group due to its interesting electronic, magnetic and optical properties, being also suitable for applications in the glass industry [27], polymers, biochemical sensors [28], and solar cells [29]. Due to its extensive development and use, terbium is inevitably present in organisms, the environment, and food chains.

Various researchers have shown that a smaller amount of doped Tb^3+^ ions can significantly increase its luminescence properties, while maintaining the main physicochemical properties and biological activities of the material [7]. Other researchers have evaluated the potential applications of HAp/Tb^3+^ in the biomedical area and their results have shown an in vitro cytocompatibility with MC3T3-E1 and A549 cells. This fluorescence property of terbium-doped HAP nanomaterials makes them suitable to be incorporated into living cells [30,31].

## 2. Experimental Section

### 2.1. Materials

All used reagents were of analytical grade and were used without further purification. Reagent-grade calcium nitrate tetrahydrate (Ca(NO_3_)_2_·4H_2_O, 99.9%, Sigma Aldrich, St. Louis, MO, USA), terbium-(III) nitrate pentahydrate (Tb(NO_3_)_3_·5H_2_O, 99.9%, Alfa Aesar, Haverhill, MA, USA), ammonium phosphate dibasic ((NH_4_)_2_HPO_4_, 99.0%, Alfa Aesar, Haverhill, MA, USA), and ammonia solution (NH_4_OH, 25% solution) were used for the synthesis. The compounds were dissolved in deionized water.

### 2.2. Synthesis Procedure

Samples Notation: HAp represents the undoped hydroxyapatite; Ca_10−x_Tb_x_(PO_4_)_6_(OH)_2_ represents the hydroxyapatite doped with different concentration of terbium ions, where x = 0, 0.05, 0.1, 0.25, 0.5 and 1.

The terbium-doped hydroxyapatite (Ca_10−x_Tb_x_(PO_4_)_6_(OH)_2_) nanomaterials were synthesized using a coprecipitation method. The samples were obtained using the following synthesis procedure: initially, a solution was obtained by dissolving an appropriate amount of Ca(NO_3_)_2_·4H_2_O with terbium-(III) nitrate pentahydrate, in deionized water, under vigorous stirring at room temperature. Subsequently, to this solution was added dropwise a solution of ammonium phosphate dibasic obtained in the same conditions. The ratio Ca/P and (Ca + Tb)/P was maintained at 1.67. The atomic ratio Tb/(Tb + Ca) was varied between 0 and 10%. The reaction was kept for 2 h under continuous stirring. Meanwhile, the resulting suspension was adjusted, and pH = 10.5 was maintained by adding NH_4_OH (25%) solution. Undoped HAp was synthesized using a similar procedure, without the addition of terbium precursor. The resulting suspensions were kept for 24 h. After maturation, the precipitates were filtered off, washed with deionized water at pH value close to 7.0 and finally dried in an air oven at 80 °C for 12 h.

### 2.3. Characterization

The structure of the obtained powders was investigated with a Nicolet iS50R spectrometer, in ATR mode at 4 cm^−1^ resolution, at room temperature. Each spectrum was measured in the range of 4000–400 cm^−1^. XRD spectra were recorded using a PANalytical Empyrean diffractometer at room temperature, with a Cu X-ray tube (λ Cu Kα1 = 1.541874 Ǻ) operating with in-line focusing, with programmable divergent slit on the incident side and a programmable anti-scatter slit mounted on the PIXcel3D detector on the diffracted side. The diffraction patterns were collected in a Bragg–Brentano geometry, with a scanning step of 0.02° and a 255 s measuring time per step. The XRD patterns were recorded in the 2θ scan ranging from 20° to 80°. The lattice parameters mean crystallites sizes and strains were calculated using the High Score Plus 3.0e software and refined by the Rietveld method. A Quanta Inspect F50 FEG (field emission gun) scanning electron microscope with 1.2 nm resolution was used for SEM examinations. The microscope was equipped with an energy-dispersive X-ray (EDX) analyzer (resolution of 133 eV at MnK_α_, FEI Company) on sample covered with a thin gold layer. The bright field TEM micrographs were obtained using a Tecnai G2 F30 S-Twin high-resolution transmission electron microscope from Thermo Fisher (former FEI) (Waltham, MA, USA), which operated at an acceleration voltage of 300 kV. An Able Jasco V-560 spectrophotometer, with a scan speed of 200 nm/s, between 200 and 850 nm, was used for obtaining the UV-Vis diffused reflectance spectra. The PL spectra were measured by using a Perkin Elmer LS 55 fluorescence spectrophotometer, with a scan speed of 200 nm/s between 350 and 800 nm, and with excitation and emission slit widths of 7 nm. The used excitation wavelength was 320 nm. The samples were finely ground by using an agate mortar and pestle. The fine powder was placed afterwards in the solid sample holder of the solid sample accessory of the device. The content of terbium in the mineral phase of the substituted hydroxyapatite was analyzed using an inductively coupled plasma mass spectrometry (ICP-MS). Prior to ICP-MS analysis (Agilent 8800, Agilent Technologies, Santa Clara, CA, USA), approximately 10 mg of sample was dissolved in 100 μL HNO3 and diluted in 10 mL volumetric flasks with ultrapure water. Solutions of 1 mg/mL Ca_10−x_Tb_x_(PO_4_)_6_(OH)_2_ (x = 0, 0.05, 0.1, 0.25, 0.5, 1) were further diluted by 1000 and 10,000 times, respectively. Calibration curves, ranging from 1 ppb to 50 ppb, were prepared using multielement standard solution (multi-element calibration standard-2A, Agilent Technologies). Linear calibrations, with a correlation coefficient greater than 0.99, were obtained for all elements.

### 2.4. Cellular Viability Assays

#### 2.4.1. MTT Assay

The biocompatibility of synthesized terbium-doped hydroxyapatite samples was estimated using MTT [3-(4,5-dimethylthiazolyl)-2,5-diphenyltetrazolium bromide] assay (Vybrant^®^MTT Cell Proliferation Assay Kit, Thermo Fischer Scientific, Waltham,, MA, USA). The human mesenchymal amniotic fluid stem cells (AFSC) were grown in DMEM medium (Sigma-Aldrich, Saint Luis, MO, USA) supplemented with 10% fetal bovine serum, 1% antibiotics (penicillin and streptomycin) (Sigma-Aldrich, Saint Luis, MO, USA), changed twice a week. The growth of AFSC cells took place in 96-well plates, with a seeding density of 3000 cells/well in the presence of Ca_10−x_Tb_x_(PO_4_)_6_(OH)_2_ powders, for 72 h. Subsequently, 15 mL (12 mM) of MTT was added to the cells, followed by incubation at 37 °C for 4 h. A solution of 1 mg Sodium Dodecyl Sulphate in 10 mL HCl (0.01 M) was added and pipetted vigorously in order to solubilize the formed formazan crystals. A TECAN Infinite M200 spectrophotometer (Männedorf, Switzerland) at 570 nm was used to evaluate the optical density (OD) of solubilized formazan, after 1 h.

#### 2.4.2. GSH-Glo Glutathione Assay

The oxidative stress assessment was performed using the GSH-Glo Glutathione Assay kit (Promega, WI, USA). AFSC was seeded at a density of 3000 cells in 300 µL of Dulbecco’s Modified Eagle’s medium (DMEM) supplemented with 10% fetal bovine serum and 1% antibiotics (penicillin, streptomycin/neomycin) in 96-well plates. After 24 h of seeding, cells were treated with Ca_10−x_Tb_x_(PO_4_)_6_(OH)_2_ and then incubated for 72 h. To this solution was added 100 µL 1X GSH-Glo Reagent, followed by incubation at 37 °C for 30 min. Subsequently, 100 µL Luciferin Detection Reagent was added and incubated at 37 °C for an additional time of 15 min. Afterwards, the medium from the wells was well homogenized and then the plate was read on the luminometer (Microplate Luminometer Centro LB 960, Berthold, Germany) [32,33,34].

#### 2.4.3. Fluorescence Microscopy

A RED CMTPX fluorophore (Thermo Fischer Scientific, Waltham,, MA, USA) was used to evaluate the biocompatibility of the obtained Ca_10−x_Tb_x_(PO_4_)_6_(OH)_2_ (x = 0, 0.05, 0.1, 0.25, 0.5, 1) powders. The CMTPX (cell tracker for long-term tracing of living cells tracker) was added to the cell culture, previously treated with the synthesized nanoparticles. The viability and morphology of the AFSC was evaluated after 5 days. To allow the dye penetration into the cells, the CMTPX fluorophore, at a concentration of 5 µM and incubated for 30 min, was added in the culture medium. Finally, the AFSC were washed with PBS and visualized by fluorescent microscopy using an Olympus CKX 41 digital camera driven by CellSense Entry software (Olympus, Tokyo, Japan) [35].

## 3. Results and Discussions

### 3.1. ICP-MS Analysis

The concentration of terbium ions from doped HAp was analyzed by ICP-MS technique. Table 1 shows the measured concentrations, the correlation coefficient, and the limit of detection.

The data from Table 1 confirm the presence of terbium ions in doped hydroxyapatite powders. It was found that the content of dopant ion increases from 0 to 1 of ions doped HAp.

### 3.2. FTIR Analysis

Figure 1 shows the FTIR spectra of Ca_10−x_Tb_x_(PO_4_)_6_(OH)_2_ samples. The absorption bands characteristic of HAp and reported in literature [36] appear in the HAp spectrum. The FTIR spectrum of pure HAp (black line in Figure 1) shows a broad band in the region 3000–3400 cm^−1^ which corresponds to the -OH groups. The bands around 1090, 1023 and 960 cm^−1^ are due to the stretching mode of P-O [37]. The bands around 602 cm^−1^, 562 cm^−1^ and 474 cm^−1^ are attributed to the bending mode of O-P-O [38]. The band at around 873 cm^−1^, which appears in all studied compound spectra, is due to the [HPO_4_]^2−^ ions [39,40].

Comparing the FTIR spectra, it can be seen that the spectra of Ca_10−x_Tb_x_(PO_4_)_6_(OH)_2_ powders with various rare earth concentrations are similar to the FTIR spectrum of pure HAp. In FTIR spectra of Ca_10−x_Tb_x_(PO_4_)_6_(OH)_2_ samples, the bands around 873 and 1426 cm^−1^ are assigned to CO_3_^2−^. This can be attributed to the CO_3_^2−^ groups replacing the PO_4_^3−^groups, signifying an interaction between HAp and carbon dioxide from air [41]. The intensity of PO_4_^3−^ bands decreases with increasing terbium concentrations until the molar fraction of terbium is 0.1%. Above this concentration, the banding bands of O-P-O increase. An explanation for this is that by replacing calcium ions with terbium ions, a change of bonding forces between the ions is induced, resulting in weakness of the banding bands of O-P-O [42]. The mechanism of substitution of Ca ions with rare earth ions has not been fully elucidated, on the basis of these studies. The increasing concentration of doped terbium ions caused a decrease in the intensity of the bands, associated with a decrease of HAp crystallinity.

### 3.3. XRD Analysis

The XRD patterns of the studied terbium-doped hydroxyapatite and undoped HAp reveal the formation of a pure hexagonal HAp phase (according to ICDD PDF4+ card no 00-068-0738, Figure 2 [43], in agreement with the literature [44,45,46]. The XRD patterns of the studied samples indicate only the pure hexagonal HAp phase of the space group P6_3_/m, along with all of the diffraction peaks in the HAp standard JCPDS database (PDF4+ card no 00-068-0738), such as: (002), (121), (112), (030), (022), (130), (222), (123) and (004).

The crystallinity degree and crystallite size of the studied samples are shown in Table 2, and an illustrative variation of those parameters is presented in Figure 2. In all of the studied samples, the peak intensities decrease with increasing terbium ion doping concentration level, signifying an interference of terbium ions with the crystal structure of HAp.

The average crystallite size of the undoped HAp was 6.07 nm. For terbium-doped HAp samples, a decrease in crystallite size was observed with increasing ion dopant content. The lattice parameters of the terbium ion-doped HAp powders slowly decreased with increasing content of the terbium cations (Table 2). The introduction of Tb^3+^ in HAp lattice induces a decrease of crystallite size from 5.72 to 4.83 nm and an increase of microstain from 1.61% to 191%. Then, for a substitution degree of more than 10%, an increase of crystallite size up to 6.43 nm was observed.

The lattice parameters *a* = *b* and *c* obtained from XRD peaks for undoped HAp have values of a = b = 9.4227 Å and c = 6.8837 Å, which is in accord with literature data [47,48]. The small changes in the lattice parameters are due to the ionic radius of terbium and calcium ions: Tb^3+^ (0.923 Å) and for Ca^2+^ is 1Å. The small differences observed for the studied samples are caused by the small differences in ionic radius [48]. When one ionic radius is greater than the others, a slow lattice expansion is observed. For terbium ions at low concentration, differences are not observed, but a higher substitution degree results in substantial changes in the unit cell parameters.

From the data in Table 3 and Figure 3 and Figure 4 show the values of the unit cell parameters *a*, *c*, *V* and the agreement indices obtained from Rietveld analysis (R_exp_, R_p_, R_wp_ and χ^2^), an indication of the quality of fit, for hydroxyapatite HAp and terbium-doped HAp with different concentrations. The evaluation of crystallinity degree indicates a decrease in crystallinity with increasing ion dopant contents for all studied samples.

### 3.4. SEM Investigation

The SEM morphologies of pure HAp and terbium-doped hydroxyapatite with various concentrations are shown in Figure 5 and Figure 6. The SEM images recorded for HAp highlight a morphology with agglomerated nanoparticles, in the form of rods, but also intergranular spaces. The obtained particles have a size in the range of 5–9 nm.

By doping with different concentrations of terbium ions, a small influence can be detected in the morphology of substituted HAp when compared to pure HAp.

Figure 6 shows SEM images of the hydroxyapatite doped with terbium of different concentrations. By doping hydroxyapatite with terbium ions, the morphology was modified. Thus, the nanoparticles become spherical, the agglomerates denser and their size is in the range of 3–7 nm. The SEM findings were supported by TEM analysis.

### 3.5. TEM Investigation

In Figure 7, TEM images of hydroxyapatite doped with terbium ion for x = 0.05 (Figure 7a), x = 0.25 (Figure 7b) and x = 1 (Figure 7c) are shown. The bright field TEM images show a rod-like morphology of the particles specific to hydroxyapatite.

The TEM micrographs confirmed what was also shown in SEM images, in terms of the tendency to form agglomerates because of the nanometric dimensions of the particles and the decrease in particle size with increasing terbium content.

### 3.6. UV-Vis and PL Spectra

The UV-Vis spectra of terbium-doped hydroxyapatite powders from Figure 8a shows two shoulders at 242 nm and 259 nm. The intensity of band observed at 242 nm increases with increasing terbium content. No other bands are observed in UV-Vis absorption spectra of terbium-doped hydroxyapatite powders.

The PL emission spectra of Ca_10−x_Tb_x_(PO_4_)_6_(OH)_2_ at different concentrations from Figure 8b present two distinct domains. The first domain, which covers a wavelength range from 350 nm to 530 nm, contains emission bands due to the emission spectrum of HAp and oxygen vacancy, interstitial or oxygen antisites. The intensity of the PL emission spectra in this domain increases with terbium content up to x = 0.05 and then decreases until x = 1. The most intense observed emission peaks are associated with: ^5^D_4_ → ^7^F_6_ (~495 nm); ^5^D_4_ → ^7^F_5_ (542 nm); ^5^D_4_ → ^7^F_4_ (583 nm); and ^5^D_4_ → ^7^F_3_ (619 nm) [48]. The intensities of these maxima increase with increasing terbium content.

### 3.7. Cell Viability and Cytotoxicity Assessment by MTT Assay, GSH Assay and Fluorescence Microscopy

MTT assay from Figure 9 shows that the obtained terbium-doped hydroxyapatite nanomaterials present absorption values close to the control sample at 72 h, proving their non-toxicological effect.

Glutathione (Figure 10) has a response for all terbium-doped nanomaterials similar to the control cells, indicating that the obtained nanomaterials do not induce cellular stress.

Fluorescence microscopy images in Figure 11 show the viability of AFSC cells; Ca_10−x_Tb_x_(PO_4_)_6_(OH)_2_ nanomaterial samples have no cytotoxic effect, confirming the biochemical results. No dead cells or cell fragments are detected; AFSCs have a normal morphology, with a distinctive appearance. 

AFSC presents extensions that suggest an active phenotype. These are possible due to the activity of the cytoskeleton and mainly represent actin filaments and microtubules. Cellular metabolism is active, as shown by fluorescence microscopy images, with cells absorbing the fluorophore CMTPX dye in the cytoplasm, suggesting their viability after five days of incubation with amniotic fluid stem cells.

## 4. Conclusions

In summary, the influence of terbium ions on the formation and structure of hydroxyapatite biomaterials synthesized by the coprecipitation method, was investigated. Different concentrations of terbium were considered, described by the formula Ca_10−x_Tb_x_(PO_4_)_6_(OH)_2_, where x = 0.05; 0.1; 0.25; 0.5 and 1. The FTIR analysis and X-ray diffraction proved the successful replacement of the calcium ions from HAp lattice with terbium ions until a sample concentration of x = 0.5 was achieved. Increasing concentration of terbium ions resulted in a decrease of HAp crystallinity. SEM and TEM analysis confirmed a spherical morphology with a tendency to form agglomerates, caused by their nanometric dimension. However, when doping of HAp is performed with different concentrations of terbium ions, it was shown to retain its physicochemical characteristics, while conferring new photoluminescence properties. The biocompatibility study of the obtained Ca_10−x_Tb_x_(PO_4_)_6_(OH)_2_ powders evaluated by fluorescence microscopy revealed a pronounced effect of cell viability and proliferation with the existence of a greater number of active cells. Therefore, the present study provides improved research consisting of obtaining terbium-doped hydroxyapatite nanomaterials with new photoluminescence properties for biomedical applications.

## Figures and Tables

**Figure 1 nanomaterials-11-02442-f001:**
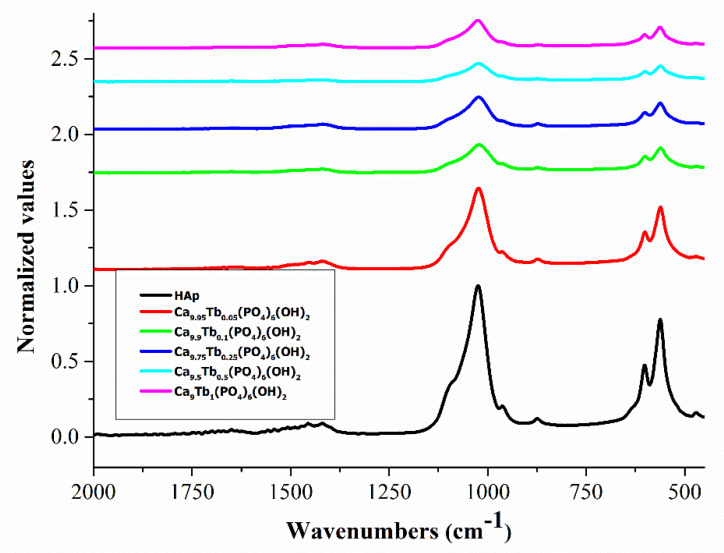
Normalized FTIR spectra of hydroxyapatite (HAp) and Ca_10−x_Tb_x_(PO_4_)_6_(OH)_2_.

**Figure 2 nanomaterials-11-02442-f002:**
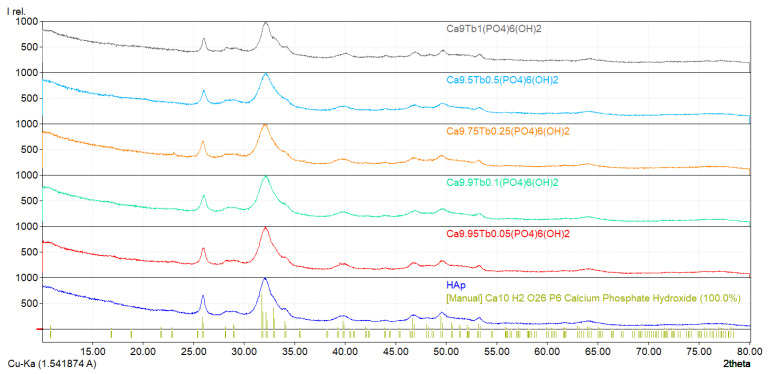
X-ray diffraction patterns of hydroxyapatite (HAp) and Ca_10−x_Tb_x_(PO_4_)_6_(OH)_2_ powders.

**Figure 3 nanomaterials-11-02442-f003:**
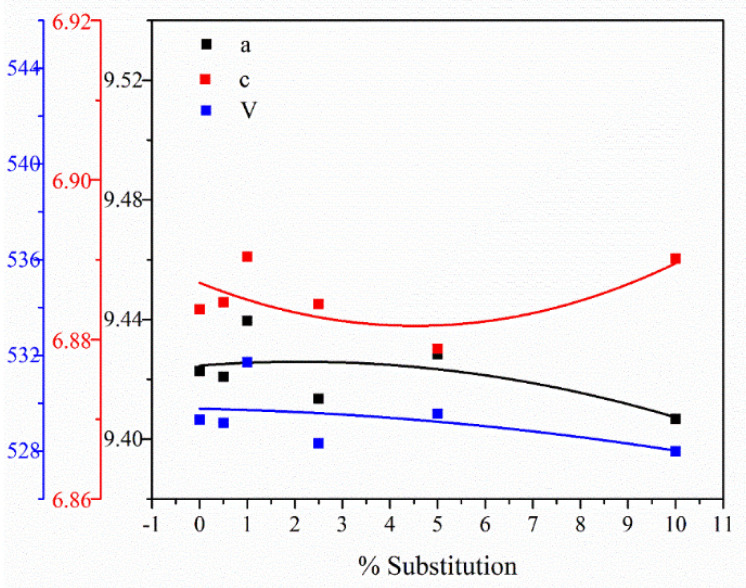
Unit cell parameters (*a*, *c*, *V*) *vs*. substitution degree for Ca_10−x_Tb_x_(PO_4_)_6_(OH)_2_.

**Figure 4 nanomaterials-11-02442-f004:**
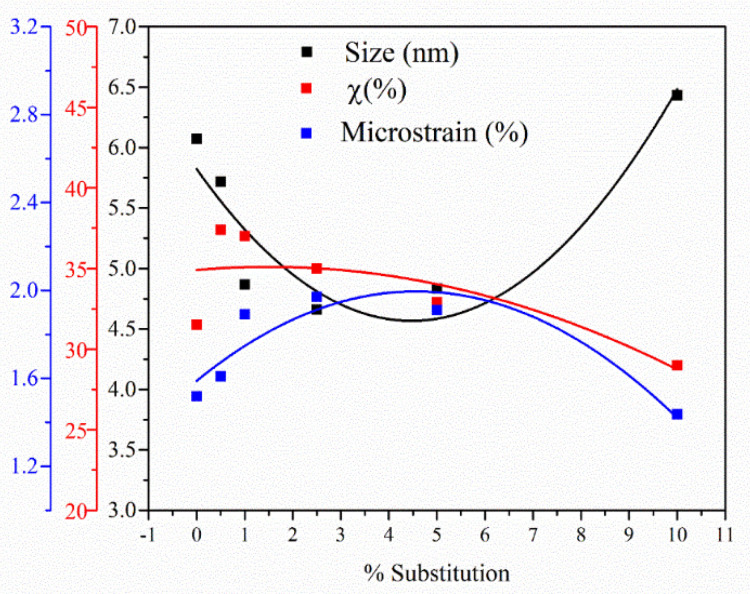
The estimated lattice microstrain, crystallite size and degree of crystallinity for Ca_10−x_Tb_x_(PO_4_)_6_(OH)_2_.

**Figure 5 nanomaterials-11-02442-f005:**
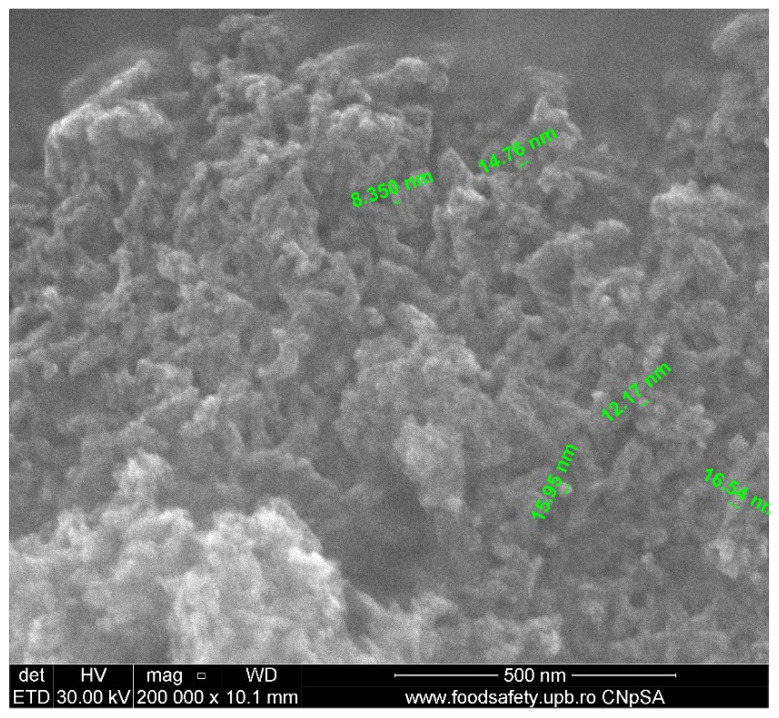
SEM image of pure HAp.

**Figure 6 nanomaterials-11-02442-f006:**
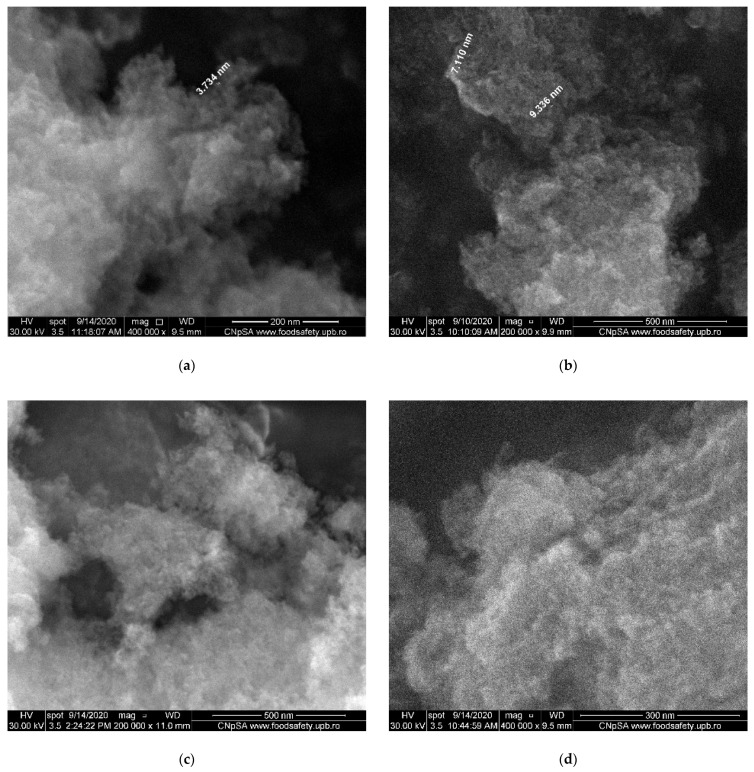
SEM images of Ca_10−x_Tb_x_(PO_4_)_6_(OH)_2_ (**a**) x = 0.05; (**b**) x = 0.1; (**c**) x = 0.25; (**d**) x = 0.5; (**e**) x = 1.

**Figure 7 nanomaterials-11-02442-f007:**
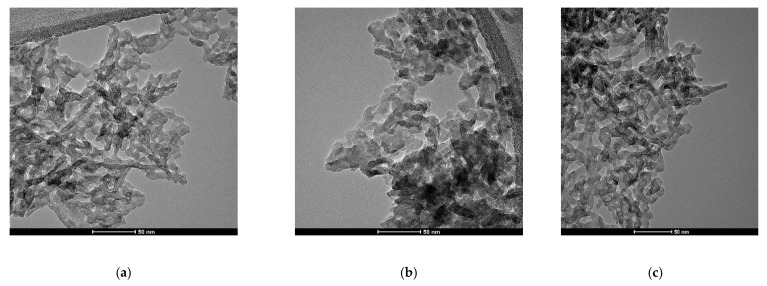
TEM images, for x = 0.05 (**a**)**,** x = 0.25 (**b**), x = 1 (**c**).

**Figure 8 nanomaterials-11-02442-f008:**
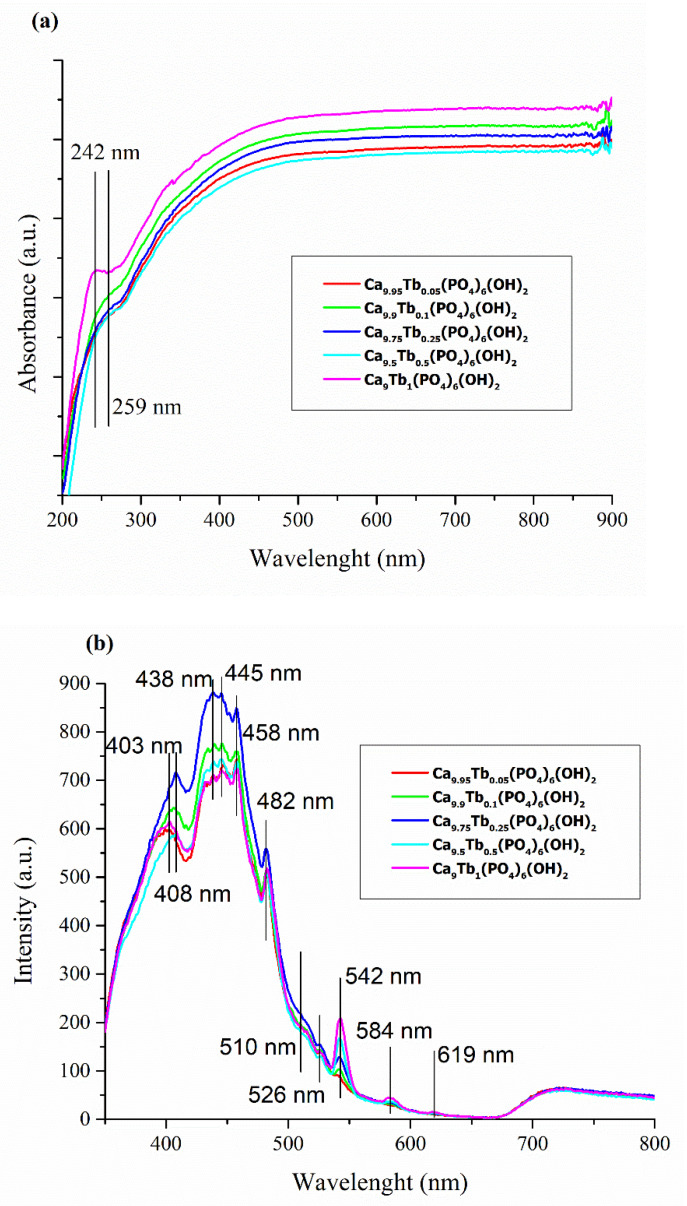
(**a**) UV-Vis absorption spectra of Ca_10−x_Tb_x_(PO_4_)_6_(OH)_2_; (**b**) PL spectra of Ca_10−x_Tb_x_(PO_4_)_6_(OH)2.

**Figure 9 nanomaterials-11-02442-f009:**
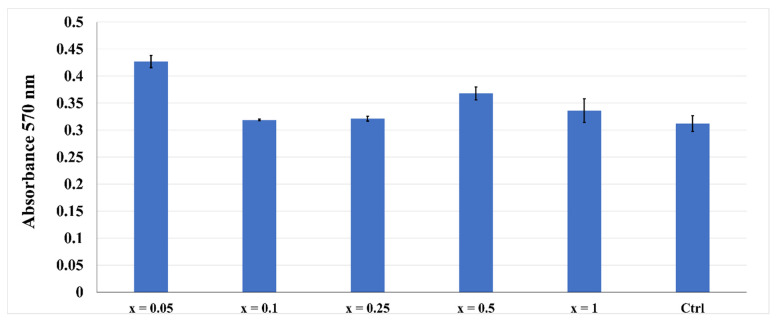
The viability of AFSC in the presence of the terbium-doped HAp by MTT assay.

**Figure 10 nanomaterials-11-02442-f010:**
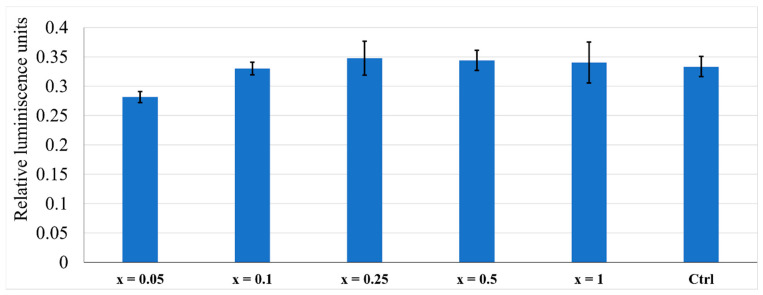
The AFSC of the terbium-doped HAp nanomaterials and control samples by GSH assay.

**Figure 11 nanomaterials-11-02442-f011:**
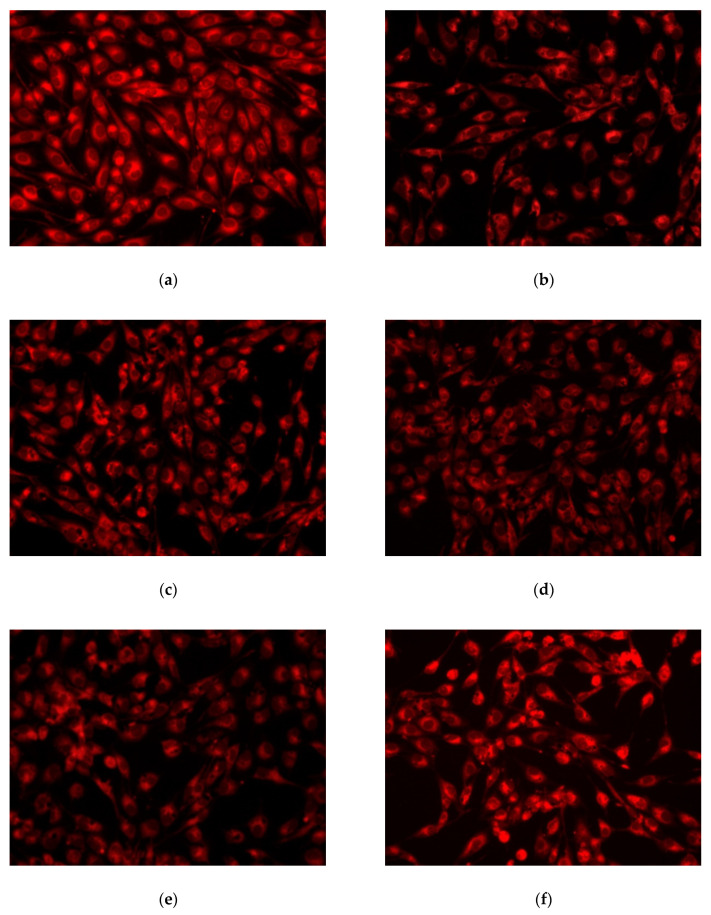
Fluorescence images of Ca_10−x_Tb_x_(PO_4_)_6_(OH)_2_ samples colored with CMTPX fluorophore: (**a**) control sample; (**b**) x = 0.05; (**c**) x = 0.1; (**d**) x = 0.25; (**e**) x = 0.5; (**f**) x = 1.

**Table 1 nanomaterials-11-02442-t001:** Terbium ion contents for substituted HAp.

Sample	Element	µg Element/mg Sample	Limit of Detection (LoD) [ug/L]	Correlation Coefficient (r^2^)
Ca_9.95_Tb_0.05_(PO_4_)_6_(OH)_2_	^159^Tb	7.89	0.0002	0.997
Ca_9.9_Tb_0.1_(PO_4_)_6_(OH)_2_		25.11		
Ca_9.75_Tb_0.25_(PO_4_)_6_(OH)_2_		41.27		
Ca_9.5_Tb_0.5_(PO_4_)_6_(OH)_2_		70.66		
Ca_9_Tb_1_(PO_4_)_6_(OH)_2_		110.37		

**Table 2 nanomaterials-11-02442-t002:** Calculated crystallite size (D) values and degree of crystallinity (χ_c_) of pure HAp and terbium-doped hydroxyapatite with various concentrations.

Samples	D/nm	S/%	χ_c_/%
HAp	6.07 ± 0.82	1.52 ± 0.53	31.50
Ca_10−x_Tb_x_(PO_4_)_6_(OH)_2_			
Ca_9.95_Tb_0.05_(PO_4_)_6_(OH)_2_	5.72 ± 0.76	1.61 ± 0.54	37.40
Ca_9.9_Tb_0.1_(PO_4_)_6_(OH)_2_	4.87 ± 0.71	1.89 ± 0.60	37.00
Ca_9.75_Tb_0.25_(PO_4_)_6_(OH)_2_	4.66 ± 0.72	1.97 ± 0.60	35.02
Ca_9.5_Tb_0.5_(PO_4_)_6_(OH)_2_	4.83 ± 0.57	1.91 ± 0.69	32.87
Ca_9_Tb_1_(PO_4_)_6_(OH)_2_	6.43 ± 0.70	1.44 ± 0.56	28.95

**Table 3 nanomaterials-11-02442-t003:** Unit cell parameters *a*, *c*, *V* and agreement indices for hydroxyapatite HAp andCa_10−x_Tb_x_(PO_4_)_6_(OH)_2_ (with x = 0.05, 0.1, 0.25, 0.5 and 1).

Sample	*a* [Å]	*c* [Å]	*V* [Å^3^]	R_exp_	R_p_	R_wp_	χ^2^
HAp	9.4227 ± 0.0033	6.8837 ± 0.0025	529.3087	3.1050	4.5059	5.7040	3.3747
	Ca_10−x_Tb_x_(PO_4_)_6_(OH)_2_						
Ca_9.95_Tb_0.05_(PO_4_)_6_(OH)_2_	9.4208 ± 0.0030	6.8846 ± 0.0023	529.1685	3.6765	3.8036	5.0633	1.8967
Ca_9.9_Tb_0.1_(PO_4_)_6_(OH)_2_	9.4395 ± 0.0037	6.8903 ± 0.0029	531.7138	3.6495	3.6622	4.7929	1.7247
Ca_9.75_Tb_0.25_(PO_4_)_6_(OH)_2_	9.4135 ± 0.0043	6.8844 ± 0.0035	528.3299	3.4967	3.4024	4.5551	1.6969
Ca_9.5_Tb_0.5_(PO_4_)_6_(OH)_2_	9.4283 ± 0.0046	6.8788 ± 0.0036	529.5652	3.3414	3.3127	4.2472	1.6157
Ca_9_Tb_1_(PO_4_)_6_(OH)_2_	9.4066 ± 0.0037	6.8901 ± 0.0028	527.9956	3.1025	2.9919	3.9236	1.5994

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
