# Peer review of "Influence of Terbium Ions and Their Concentration on the Photoluminescence Properties of Hydroxyapatite for Biomedical Applications"

_nanomaterials, 2021, doi:10.3390/nano11092442_

Round 1

Reviewer 1 Report

The paper presents a detailed description of the influence of terbium ions on the formation and structure of hydrox-yapatite biomaterials, synthesized by co-precipitation method. Although the paper is well organized and the results appear to be solid and interesting, I have found some passages difficult to read properly. This difficulty could be due to a too "dry and direct" description of the work done. The authors have published several works on very upcoming materials, and I suggest them giving a greater overview of the topics described. For example, the abstract should be reorganized e reformulated not only limiting to the results presented. 

In addition, figure 1 presents the relevant results in the range 500-1500, the portion 2000-4000 can be omitted, or if included in the plot, the authors should justify if peaks are expected in such range.

The overall presentation must be improved.

The English presents some grammar errors and several sentences should be rephrased, I suggest having a native speaker read the paper.

Reviewer 2 Report

The paper deals with the doping of Ca10(PO4)6(OH)2 by luminescent trivalent Tb3+ ions.

The FTIR characterization is not clear:

  • The authors state “The FTIR spectrum of pure HAp (black line from Figure 1) shows a broad band in the region 3000 - 3400 cm-1 which corresponds to adsorbed water”. As hydroxyapatite contains -OH groups, I would expect this broad band to be assigned to the -OH groups.
  • The graph should be zoomed so that the bands discussed in the text (around 600 cm-1) can be clearly visualized.
  • The authors state “the banding bands of O-P-O increases”. How do they determine that the bands increase? Do they normalize all the spectra? Do they compare relative band intensity? If not, the spectra cannot be compared in intensity as FTIR is not purely quantitative.

Regarding the XRD diagrams (Figure 2), the authors should add the reference diagram on their experimental data so that the reader can assess from the phase purity.

Still on the same graph, the black diagram is much more intense than the others. The authors should normalize them for better visualization.

The authors discuss the degree of crystallinity but they do not explain how it is calculated. In particular, the black diagram exhibits a strong amorphous background (at low angles) but the authors report a crystallinity degree similar to the other samples. How do they explain this phenomenon?

Similarly, the authors report strain and crystallite size values coming from Rietveld fit. They should provide an example of their Rietveld fitting and the R2 value.

The fits proposed on Figure 3 are not convincing.

Similarly, the SEM images (Figure 5 and 6) are not convincing as it is highly difficult to identify particles with a size smaller than 10 nm. I suggest the authors should perform TEM experiments on the different samples to study the influence of the doping on the particle morphology.

Figure 8a reports a negative absorbance, A, which is not possible as A =log (I0/I). It probably comes from an experimental artefact.

The caption of Figure 8b is missing.

Figure 8b shows the PL spectra of different samples. Comparison of PL intensity is very complicated when performed on powder samples because of scattering bias and sample preparation. The authors should explain how the measurements are carried out if they want to compare the different sample PL intensity.

The conclusion is inconsistent as the authors say: “the successfully replacement of the calcium ions from HAp lattice by terbium ions until the sample concentration with x = 0.5.” and then “the 10% degree of doping is the limit of terbium ions which are accepted in the hydroxyapatite lattice.”

Author Response

Please see atachment.

Reviewer 3 Report

The authors investigated the influence of Tb ions on photoluminescence properties of hydoxyapatite for possible bioimiging.. The manuscript was supplied with all data needed to repeat the syntheses. The experimental techniques used are adequate. The biocompability and cytotoxicity of prepared  samples were tested.

I do recommend this manuscript as publication.

Round 2

Reviewer 2 Report

-